# SYNTHETIC REDUCED NEAREST NEIGHBOR MODEL FOR REGRESSION

## ABSTRACT

Nearest neighbor models are among the most established and accurate approaches to machine learning. In this paper, we investigate Synthetic Reduced Nearest Neighbor (SRNN) as a novel approach to regression tasks. Existing prototype nearest neighbor models are initialized by training a k-means model over each class. However, such initialization is only applicable to classification tasks. In this work, we propose a novel initialization and expectation maximization approach for enabling the application of SRNN to regression. The proposed initialization approach is based on applying the k-means algorithm on the target responses of samples to create various clusters of targets. This is proceeded by learning several centroids in the input space for each cluster found over the targets. Essentially, the initialization consists of finding target clusters and running k-means in the space of feature vectors for the corresponding target cluster. The optimization procedure consists of applying an expectation maximization approach similar to the k-means algorithm that optimizes the centroids in the input space. This algorithm is comprised of two steps: (1) The assignment step, where assignments of the samples to each centroid is found and the target response (i.e., prediction) of each centroid is determined; and (2) the update/centroid step, where each centroid is updated such that the loss function of the entire model is minimized. We will show that the centroid step operates over all samples via solving a weighted binary classification. However, the centroid step is NP-hard and no surrogate objective function exists for solving this problem. Therefore, a new surrogate is proposed to approximate the solution for the centroid step. Furthermore, we consider the consistency of the model, and show that the model is consistent under mild assumptions. The bias-variance relationship in this model is also discussed. We report the empirical evaluation of the proposed SRNN regression model in comparison to several state-of-the-art techniques.

## 1 INTRODUCTION

One of the main topics of research in Machine Learning is the relation between the features and output responses Hastie et al. (2009); Santosa & Symes (1986); Tibshirani (1996); Criminisi & Shotton (2013). Synthetic Reduced Nearest Neighbor (SRNN) models are shown to be an effective tool in determining the relationships between features of the inputs and the sub-clusters of each class in supervised learning tasks Tavallali et al. (2020b). However, existing prototype nearest neighbor models such as SRNN are constrained to classification problems, and to the best of our knowledge, there remains a gap in extending these algorithms towards regression tasks. Such regression reduced nearest neighbor models may find extensive applications in epidemiological studies Tavallali et al. (2020a); Cisneros et al. (2021), medical studies Criminisi & Shotton (2013); Graf et al. (2011b;a), and other applied regression tasks in general Tibshirani (1996). To address this gap in the state of the art, we propose a novel algorithm for the optimization and construction of Regression Synthetic Reduced Nearest Neighbor (Reg-SRNN) models.

The proposed Reg-SRNN is capable of discovering various modalities of the input data, and relates those to the modalities of the output responses. The Reg-SRNN algorithm is designed to handle both single-response and multi-response regression. The multi-response regression consists of learning the relation between input samples and several ground-truth output responses. Reg-SRNN partitions the input space into piecewise constant regions, where each region is represented by a centroid and

its output response. From this perspective, Reg-SRNN is similar to other piecewise constant models, such as Li & Martin (2017); Begon et al. (2017); Bertsimas & Dunn (2017); Tavallali et al. (2019; 2020c). Reg-SRNN is capable of learning an accurate relation between each cluster of the data and its corresponding output responses. Therefore, Reg-SRNN can also provide enhanced interpretablity by reducing the information content of clusters into a compressed representation manifested in their centroids.

The technical contributions of this paper include the proposal of a novel initialization that by itself is competitive to other existing regression models. This is proceeded by an expectation maximization algorithm for directly minimizing the least squares error of the mode. The proposed optimization algorithm is provably convergent, and it is shown that it monotonically decreases the loss function. Therefore, the algorithm has a convergence guarantee on minimizing the loss function and achieving a local optimum.It is also worth mentioning that the algorithm does not cycle. The proposed optimization algorithm consists of two steps and is inspired by K-means algorithm Lloyd (1982). One step is the assignment step and is composed of finding samples assignments and proper output response of the centroid. Second step is the update step where the centroid is optimized such that the loss function is decreased. The centroid step is affected by all the samples and we will show that this update step is a kind of NP-hard weighted binary classification problem. The update step is computed through a surrogate objective function that is similar to SVM. We establish that the algorithm is efficient because of its linear computational complexity. Finally, the model is evaluated on various datasets with various sizes and dimensionalities, the results of which demonstrate that Reg-SRNN is capable of competing and even over-matching similar regression models.

Accordingly, the main contributions of this paper are as follows:

- We propose a novel algorithm for initialization of SRNN models to extend their application to regression tasks.
- We develop an optimization algorithm for regression SRNN models with guarantees on convergence.
- Through experimental evaluation, we demonstrated the feasibility of our proposed regression SRNN model in filling the gap between more complex models (such as random forests) and basic and interpretable models such as linear regression and decision trees.

## 2 RELATED WORK

A regression task consists of learning the relation between samples of the input space and a numerical output space. More specifically, regression is a supervised learning task of mapping inputs (independent variable $X$) to the output $Y$, which is a continuous vector ($Y \in \mathbb{R}^d$). If the dimensionality of the output $d \geq 2$, the task is known as multi-response regression. Regression has been the workhorse of numerous fields Tai (2021), and various regression models have been developed and expanded fundamentally over the recent decades Hastie et al. (2009). This expansion has been so rampant such that listing all such models and their relationships is a difficult task and is out of the scope of this work. However, a brief review of the recent models is presented in this paper. A common objective function for regression is to minimize the least squares error:

$$||\hat{Y} - Y||^2 \tag{1}$$

Where $\hat{Y}$ is the prediction. According to the Gauss-Markov theorem Gauss (1823), the least squares error can be an unbiased linear model of minimum variance of the data under certain assumptions. Ordinary least squares may fail to properly predict outcomes if it is applied to settings where the Gauss-Markov assumptions are not held. Therefore, it is important to understand the assumptions and occasionally apply the proper changes to the objective function of equation 1 to modify the model Tai (2021). Manifestations of such changes include imposing regulations or constraints over the objective function. The literature on ordinary least squares estimation has extensively dealt with some of the well-known concerns that might violate the assumptions, such as Ridge Hoerl & Kennard (1970a;b), Lasso Tibshirani (1996), Elastic Net Zou & Hastie (2005), trees Quinlan (2014), forest Breiman (2001), boosting Bühlmann & Yu (2003) and others.

Common regression models include bagging, boosting, random forest Criminisi & Shotton (2013), oblique trees Murthy et al. (1994); Norouzi et al. (2015); Heath et al. (1993), and regression SVM

Drucker et al. (1997). In the context of regression trees, various approaches of inducing a tree are presented in the literature. Most decision tree induction methods are concentrated on the splitting criterion used at the growing phase of the tree Ikonomovska et al. (2011); Levatić et al. (2014). Application of decision tree algorithms to multi-response regression has been previously considered in the literature Breiman et al. (1984); De'Ath (2002). In Breiman et al. (1984); Quinlan (1986), authors consider training a decision tree for each individual output response. However, such approach constructs a large model specially if the number of output responses are high. Another approach proposed in De'Ath (2002) consists of constructing a single decision tree for all the output responses. In other words, the model predicts all the output values simultaneously through a single decision tree. However, a model for all the outputs might not be sufficient Kocev et al. (2009) because they train model for single response rather than the true problem which is a multi-response regression. Authors in Kocev et al. (2009) have explored two approaches to the multi-response regression problem by comparing learning a model for each output separately (i.e., multiple regression trees), and learning one model for all outputs simultaneously (i.e., a single multi-target regression tree). In order to improve predictive performance, Kocev et al. (2013) has also considered two ensemble learning techniques, namely, bagging Breiman (1996); Liang et al. (2011) and random forests Breiman (2001) for regression trees and multi-target regression trees. The results showed that multi-target regression trees created more accurate and compact models.

A related topic to the problem of this paper is nearest neighbor regression. Nearest neighbor regression and local estimators are well-established methods in the literature of ordinary univariate location estimators (Benedetti (1977); Stone (1974); Tukey et al. (1977). However, as per our extensive search, there remains a gap in prototype nearest neighbor approaches to regression. The only work that considered a similar model and optimization to SRNN was Huang (2017). However, the proposed algorithm does not have guarantee of convergence or achieving some sort of optimum solution.

## 3 PROPOSED METHOD

### 3.1 PRELIMINARIES

Assume a dataset consisting of tuples $(x_i, y_i)$ where $x$, $y$ and $i$ represent input features, output responses and index number. Each tuple represent a data $x_i$ and its corresponding output response $y_i$. Here, $x_i \in \mathbb{R}^D$ and $y_i \in \mathbb{R}^d$. The Regression Synthetic Reduced Nearest Neighbor (Reg. SRNN) consists of $K$ tuples of synthetically produced centroids/prototypes $(c_j, \hat{y}_j)$ where $c$, $\hat{y}_j$ and $j$ represent the centroid's point in the input space, output prediction and index. At the inference time, the Reg. SRNN operates like a nearest neighbor model where the centroids are used as the samples. The problem of training Reg. SRNN is as follows:

$$\min_{\{(c_j,\hat{y}_j)\}_1^K} \sum_{i=1}^{N} ||y_i - \hat{y}_{j_i^*}||^2$$
$$\text{s.t} \quad j_i^* = \operatorname*{argmin}_{\{j\}_1^K} \quad d(x_i - c_j) \tag{2}$$

where $d(.)$ is a distance metric. Through this paper we use the l-2 norm as the distance metric:

$$d(x_i - c_j) = \sqrt{||x_i - c_j||^2} \tag{3}$$

Essentially the prediction of the model consists of the output prediction of closest centroid to the input sample. Officially, we define Reg. SRNN as follows:

$$NN(x) = \sum_{j=1}^{K} y_j I(x \in R_j) \tag{4}$$

Where, $NN(.)$ represents a nearest neighbor function of the $K$ centroids. $I(.)$ is an indicator function that produces 1 if the input $x$ is in the region of $R_j$. $R_j$ represents the region where the closest centroid to the points in that region is $c_j$.

## 3.2 INITIALIZATION

Numerical optimization algorithms require initialization (cite num opt). In this paper, we propose a novel initialization for the regression SRNN. Previous work on the initialization of SRNN models consisted of learning a K-means model for each class of the data. For example, in case of $M$ classes and $K$ centroids, $\frac{K}{M}$ centroids are learned for each class as initialization of the SRNN Kusner et al. (2014); Wang et al. (2016); Zhong et al. (2017); Tavallali et al. (2020b). However, such approach is not applicable to the regression. This initialization also have close ties with naive Bayes and density estimation Silverman (2018). Here, we expand this initialization to the case of Reg. SRNN.

Intuitively, the output responses can consist of several modalities. In other words, it is possible that the output responses are generated from several distributions. The clusters of such distributions can be approximated by running a K-means over the output space ($M$ centroids). Assume that $S_m$ represents the set of samples assigned to each output cluster. Next step consists of learning $\frac{KN}{|S_m|}$ centroids over the input features of the $S_m$ for all $M$ clusters. In other words, we learn centroids over the input features of each output cluster relative to the population of that cluster. The found centroids at the second step are used as initialization for the Reg. SRNN. At this step, $\hat{y}_j$ is found using the following formula:

$$\hat{y}_j = mean(y_i \in S_j) \tag{5}$$

where $S_j$ represents the set of samples that are assigned to $j^{th}$ centroid. $S_j$ essentially consists of samples where $j^{th}$ centroid is the closest centroid to them. $mean(.)$ represents the average of its input set. Note that $S_j \in R_j$.

## 3.3 CONSISTENCY

Here we discuss the consistency of the Reg. SRNN. We show that Reg. SRNN is a nonparametric and consistent under mild assumptions for continuous features. Assume an independent identically distributed (iid) dataset where it is being generated from $f(x)$. $f(x)$ represents the true function for relation between the inputs and outputs. Also assume that $f(x)$ is a piecewise constant function. Over a set of $N$ observations, consistency of a nonparametric estimator $\hat{f}_N(x)$ (such as Reg. SRNN) is shown using the following formula Parzen (1962):

$$Pr(\lim_{N \to \infty} \int_x (\hat{f}_N(x) - f(x))^2 dx = 0) = 1 \tag{6}$$

The proof of consistency of Reg. SRNN is similar to proof of consistency for regression trees Breiman et al. (1984) and density estimators in Ram & Gray (2011).

**Theorem 1** (Consistency of Reg. SRNN). *The estimator defined in equation 4 satisfies equation equation 6.*

*Proof.* Assume $B$ and $d_1$ denote the collection of all sets $t \subset X$ and a fixed positive integer, respectively. Assume that $B$ describes the sets that are the solution to a system of $d_1$ inequalities $b^T x \leq c$ where $b \in \mathbb{R}^d$ and $c \in \mathbb{R}$. Every region in the Reg. SRNN in formula equation 4 can be seen as a solution of a system of $d_1$ inequalities of the form $b^T x \leq c$ where $b$ is a hot-one-vector (only one element of $b$ is 1 and the rest are 0). Therefore, $Reg.SRNN \subset B$.

Assume a random point $X_n$ from function $f$ on $X$, ($n \geq 1$). $\hat{F}_N$ represents the empirical function learned by Reg. SRNN over $X_n$ For $N \geq 1$, $1 \leq n \leq N$, and defined on a set $t \subset X$ by

$$\hat{F}_N(t) = \frac{1}{N} \sum_{n=1}^N y_n I(X_n \in t) = mean(y_n \in R_t) = \int_t \hat{f}_N(x) dx \tag{7}$$

where $y_n = f(x_n)$ and $\hat{f}(x)$ is the estimator presented in equation 4. Using a general version of Glivenko-Cantelli theorem Vapnik & Chervonenkis (2015)

$$Pr(\lim_{N \to \infty} \sup_{t \in B} |\hat{F}_N(t) - \int_t f(x) dx| = 0) = 1 \tag{8}$$

By replacing equation equation 7 in equation 8, we have

$$Pr(\lim_{N\to\infty}\sup_{t\in B}|\int_t \hat{f}_N(x)dx - \int_t f(x)dx| = 0) = 1 \tag{9}$$

then:

$$Pr(\lim_{N\to\infty}\sup_{t\in B}\int_t |\hat{f}_N(x) - f(x)|dx \geq 0) = 1 \tag{10}$$

Further, by assuming that the region of $t$ leans toward 0 as $N \to \infty$ ($Pr(\lim_{N\to\infty}\int_t dx = 0) = 1$). Rest of the steps will follow similar to theorem 1 in Ram & Gray (2011) and we get

$$Pr(\lim_{N\to\infty}\int_x (\hat{f}_N(x) - f(x))^2 dx = 0) = 1 \tag{11}$$

$\square$

Please note that the steps are similarly done in Ram & Gray (2011) except the step of equation 7 which is different.

The consistency shows that as the number of samples go to infinity and as $\frac{K}{|S_j|} \to 0$, then the Reg. SRNN is consistent. This is provable thanks to the assumption that the true function that relates the inputs to the outputs is a piecewise constant function. This essentially means that the SRNN itself is consistent also for classification tasks.

## 3.4 THE EXPECTATION MAXIMIZATION OF REG. SRNN

The problem equation 2 represents the training problem of Reg. SRNN. SRNN generally is known as prototype nearest neighbor in other papers of literature and centroids are also called prototypes. Problem equation 2 resembles K-means Lloyd (1982) problem except that the loss function is $||x_i - c_j||$. The expectation maximization in this paper follows same approach as to K-means algorithm and the approach in Tavallali et al. (2020b). The optimization consists of two steps, assignment step and the update step.

### 3.4.1 ASSIGNMENT STEP

The assignment step consists of calculating the assignment of the samples to each centroid and finding the optimum output prediction of each centroid. The problem of this step is as follows:

$$\min_{\{\hat{y}_j\}_1^K} \sum_{i=1}^N ||y_i - \hat{y_j}^*||^2.$$
$$\text{s.t} \quad j_i^* = \underset{\{j\}_1^K}{\operatorname{argmin}} \quad d(x_i - c_j). \tag{12}$$

Note that problem equation 12 only tends to optimize over $\hat{y}_j$ for $j = 1...K$. Using sets $S_j$, problem equation 12 can be simplified to:

$$\min_{\{\hat{y}_j\}_1^K} \sum_{j=1}^K \sum_{\{i|(x_i,y_i)\in S_j\}}^N ||y_i - \hat{y}_j||^2. \tag{13}$$

The problem equation 13 can be separated over each centroid and its corresponding set. This can be done since the regions are distinct and samples can not be shared among the regions. The prediction for each region is the label of $j^{th}$ centroid $\hat{y}_j$ (the centroid that represents the region). As a result, the problem of optimizing label for each region is

$$\min_{\hat{y}_j} \sum_{\{i|(x_i,y_i)\in S_j\}}^N ||y_i - \hat{y}_j||^2. \tag{14}$$

whose minimum is presented in formula equation 5. In other words, the optimum of $\hat{y}_j$ is the mean of response of samples in the $R_j$.

As a result, in the assignment step, the $S_j$ and $\hat{y}_j$ have to be calculated for all the samples.

### 3.4.2 UPDATE STEP

The update step consists of updating the position of centroids such that the objective function in equation 2 is minimized Lloyd (1982). At this step the output prediction of the centroids are kept constant. The update step is affected by all the samples in the dataset because by changing the position of centroid, the assignments of the samples can get changed and as a result the prediction for each train sample gets changed. Further it will be shown that finding optimum of the problem in this step is NP-hard. Therefore, the centroid problem is approximated through a novel surrogate objective function.

Each centroid is optimized individually Tavallali et al. (2020b). The optimization problem for $k^{th}$ centroid consists of moving the $k^{th}$ centroid such that the samples' assignment are changed in favor of decreasing the objective function equation 2. The centroid problem is

$$
\min_{c_k} \quad \sum_{i=1}^{N} ||y_i - \hat{y_j}^*||^2.
$$
$$
\text{s.t} \quad j_i^* = \operatorname*{argmin}_{\{j\}_1^K} \quad d(x_i - c_j).
$$
(15)

Note that the optimization is only over $c_k$. We rewrite the problem equation 15 over the assignment of a sample to $k^{th}$ centroid or to the rest of centroids. For simplicity we introduce the notation $r_{ij} = d(x_i - c_j)$. The problem is

$$
\min_{c_j} \quad \sum_{i=1}^{N} ||y_i - \hat{y}_k||^2 U(r_{ij_i^*} - r_{ik}) + ||y_i - \hat{y}_{j_i^*}||^2 U(r_{ik} - r_{ij_i^*}).
$$
$$
\text{s.t} \quad j_i^* = \operatorname*{argmin}_{\{j\}_1^K, j \neq k} \quad d(x_i - c_j).
$$
(16)

In problem equation 16, $U(.)$ is a step function where it outputs $1$ if the input is larger than $0$ and otherwise it will produce $0$. Note that the input arguments of the step functions are negative of each other. This means a sample has to either get assigned to the $k^{th}$ centroid or the rest of centroids. The sample can not get assigned to both terms of equation 16. The assignment of $i^{th}$ sample to each term will produce a continuous error. This essentially means that the problem equation 16 is a weighted binary classification problem. The problem equation 16 encourages the sample to get assigned to the side that produces lower error. For simplicity we introduce the sets $S_c$ and $S_{c_k'}$. The $S_{c_k}$ represents the set of samples that produce lower error if assigned to the $c_k$ centroid and $S_{c_k'}$ represents the set of other samples. let $t_i = abs(||y_i - \hat{y}_k||^2 - ||y_i - \hat{y}_{j_i^*}||^2)$ where $abs(.)$ returns the absolute value of its input. The problem of equation 16 is equivalent to

$$
\min_{c_j} \quad \sum_{(x_i, y_i) \in S_c} t_i U(r_{ij_i^*} - r_{ik}) + \sum_{(x_i, y_i) \in S_{c_k'}} t_i U(r_{ik} - r_{ij_i^*})
$$
(17)

The problem equation 17 is a NP-hard problem Nguyen & Sanner (2013). Therefore, inspired by the SVM, we approximate the solution to problem equation 17 using the following surrogate objective function

$$
c_k^*(\mu) = \operatorname*{argmin}_{c_k} \quad \sum_{x_i \in S_c} t_i r_{ik} + \sum_{x_i \in S_{c_k'}} t_i relu(\mu r_{ij^*} - r_{ik})
$$
(18)

where $\mu$ is a penalty coefficient. Intuitively, the surrogate objective function encourages the $k^{th}$ centroid to stay close to samples of $S_c$ while staying away from samples of $S_{c_k'}$. $\mu$ is increased from $0$ to $1$ and along this path, the $c_k^*(\mu)$ that produces the smallest error for equation 17 is selected. This surrogate is a modified version of surrogate objective function in Tavallali et al. (2020b). The $\mu$ acts similar to slack variables in a SVM problem. The objective function of problem equation 18 is a continuous function; thus, a local optimum of the problem can be found using gradient-based algorithms.

### 3.5 RELATION TO EM ALGORITHM

The proposed algorithm is originally inspired by the EM algorithm used for K-means. The assignment step consists of finding the samples assigned to each centroid and finding the optimal output

prediction of each centroid. The assignment of samples is the same as calculating the prior probabilities in an EM algorithm. Finding optimum output prediction of each centroid can also be considered as a part of maximization step.

In the update step, the centroids have to be updated. At this step, the outcome of assigning each sample to each centroid is evaluated ($||y_i - y_j||^2 {}_{j=1}^{K} \forall i = 1, 2, ..., N$). This evaluation is equivalent to calculating the posterior probabilities. Then the centroid problem is approximated using these outcomes which is the maximization step.

### 3.6 Relation of Surrogate to SVM

The problem of finding the best centroid that is closer to samples of $S_c$ than any other centroid can be cast as a feasibility problem.

$$\begin{aligned}
\text{find} \quad & c_k \\
\text{s.t} \quad & r_{ik} < r_{ij_i^*} \quad \forall (x_i, y_i) \in S_c \\
& r_{ik} > r_{ij_i^*} \quad \forall (x_i, y_i) \in S_{c_k'}
\end{aligned} \tag{19}$$

However, this feasibility problem is NP-hard since the the second set of constraints are concave Sahni (1974). These constraints make the problem different from similar SVM problems where the constraints are convex and global solution can be found in efficient time.

### 3.7 Properties of the Algorithm

**computational complexity** Optimizing the centroid problem takes $\mathcal{O}(ND)$ since it uses a gradient based algorithm for solving the surrogate objective. All the $K$ centroids have to optimized at each iteration. Therefore, the computational complexity of the algorithm is $\mathcal{O}(\alpha NDK)$ where $\alpha$ is the number of iterations.

**Convergence** The convergence to a local optimum is similar to that of the K-means algorithm Lloyd (1982); Tavallali et al. (2020b). At each iteration, the error decreases and the objective function is bounded by 0 from bellow. Further, the different combination of assignment of the samples to the centroids is finite. Therefore, the algorithm stops after a finite number of iterations.

## 4 Experimental Results

In this section, the experimental results are presented to demonstrate the merits of proposed proposed algorithm. Various datasets are downloaded and used for evaluation from UCI repository Dheeru & Karra Taniskidou (2017). We partitioned the datasets using the following setups: 1- if the dataset contained a separate test set, the dataset was used as provided in the repository. In case cross-validation was needed (for specific models) and validation set was not provided then the we partitioned the trainset to 80% train and 20% validation sets. 2- if the test set was not provided we partitioned the dataset to 80% train and 20% test set. For models that needed validationset, we divided the trainset to 80% train and 20% validation set. 3- if all sets were provided by the repository, then the sets were used as provided. The setup for Reg. SRNN is as follows: at the initialization phase, we used $K = 4$ for the output clusters for slice localization data and for the rest of datasets $K = 2$ was used for the output clusters. The number of input centroids used for each output cluster were proportional to the population of each cluster. The centroid problem was optimized using batch stochastic gradient descent. We used various setups for each dataset to achieve the smallest possible train error.

Various models are used for comparison with Reg. SRNN. The basis of comparison is based on the number of base models used in each model. All the models comparable to the Reg. SRNN can be seen as a kind of ensemble model that consists of several base models. In case of forest and boosting, each tree is a base model. For prototype models such as Reg. SRNN and K-means, each centroid is a base model. For Radial Basis Function (RBF) models, each RBF is considered as a base model. We also compared our model with linear regression and Ridge regression. The two models are presented with straight lines. Figures 1, and 2 present performance of each model

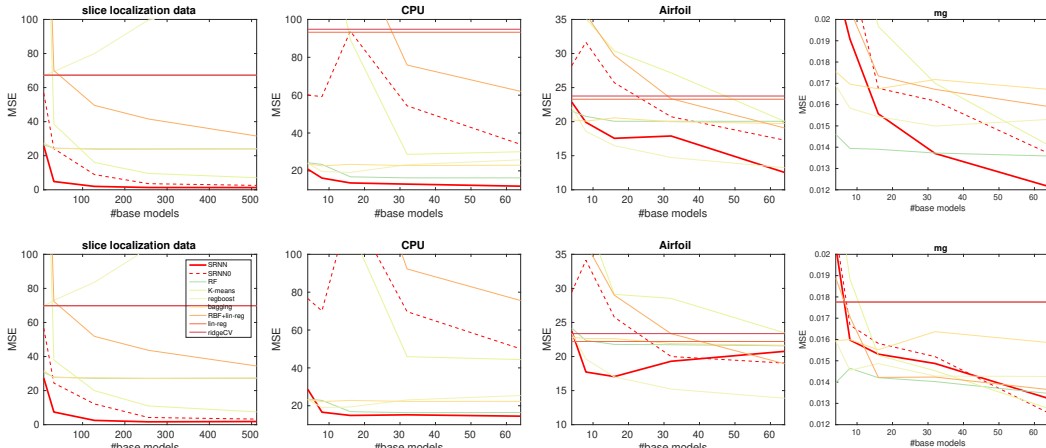

Figure 1: Horizontal axis represents the number of base models(e.g., centroids, RBFs, trees and etc.). The vertical axis represents the mean squared error. The first and second row curves show the train errors and test errors, respectively. Reg. SRNN is compared with similar models.

over various datasets based on mean squared error versus number of base models. Reg. SRNN was compared with Random Forest (RF), K-means, Regression Boosting (regboost), bagging, Radial Basis Function+ linear regression (RBF+lin-reg), linear regression (lin-reg), and ridge regression (ridgeCV). For the RF, each tree was trained using $70\%$ of the trainset and $0.7$ of features were randomly selected and used at the split nodes. For the K-means, a K-means model with desired number of centroids were trained over trainset and the assignment step of Reg. SRNN was applied to the model. For the regboost, we used trees of depth 3. For bagging, similar setup to RF was used except that no feature randomization is applied. RBF+lin-reg consists of first training RBFs (same number of RBFs as the centroids in other models) and the training linear regression over the outputs of RBFs. The width of RBFs were selected by cross-validation. The penalty coefficient of ridgeCV was selected by cross-validation.

As can be observed from figures 1, and 2, the SRNN-Reg was able to achieve better or comparable train and test errors to other models.

## 5 LIMITATIONS:

The model presented in this paper is new in the sense that it has never been proposed previously. However, since its nature is a nearest neighbor model we expect that the model presents some limitations similar to those of a nearest neighbor model. Similar to nearest neighbor for regression that might have high test error in high dimensional datasets Hastie et al. (2009). Therefore, we suspect that the performance of Reg-SRNN might deteriorate with higher dimensional dataset. However, in our experiments, we noticed that the Reg-SRNN performed very well in slice localization data which is a high dimensional dataset (384 features).

## 6 CONCLUSION

In this paper, as per our search, we have proposed the first regression synthetic reduced nearest neighbor. The consistency of the algorithm was proved and its properties were explored. We showed that the algorithm is computationally efficient and can converge to a local optimum in the sense than no move can improve the model any further. The approach is essentially the same type of EM algorithm used for K-means and it is extended for the case of regression synthetic reduced nearest neighbor. The update step was an NP-hard weighted binary classification problem. The optimum of such problems are typically approximated using a surrogate objective function, such as hinge loss in SVM for 0-1 binary classification problem. Therefore, we approximated the solution to the update step through a novel surrogate objective function. Further, we analyzed the relation of the update

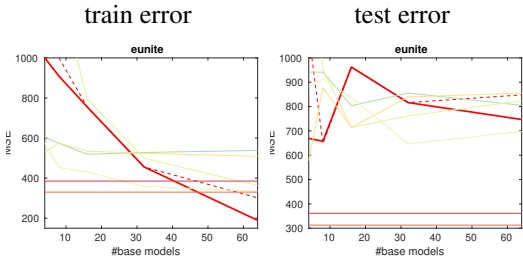

Figure 2: Horizontal axis represents the number of base models(e.g., centroids, RBFs, trees and etc.). The vertical axis represents the mean squared error. Reg. SRNN is compared with similar models.

step with SVM. Experimentally, we showed that the Reg. SRNN performs better or competitive to other similar models in the literature such as ensembles and centroid based models.

## 7 REPRODUCIBILITY

The experiment codes (currently attached as supplementary materials) will be published on a Github repository to facilitate reproduction and extension of our results. Furthermore, the datasets used in our experiments are publicly available via the UCI repository.

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
