# OpenReview forum: "Synthetic Reduced Nearest Neighbor Model for Regression"
_ICLR.cc/2022/Conference — ICLR 2022 Submitted_

### Official Review · Reviewer_j59H · 2021-10-27

**Correctness:** 3
**Technical Novelty And Significance:** 3
**Empirical Novelty And Significance:** 3
**Recommendation:** 5
**Confidence:** 4

**Main Review:**

[COMMENTS AFTER REBUTTAL]

I have read the rebuttals from authors and reviews from other reviewers. However, I still think that this submission is below the standard I expect for an ICLR paper. Thanks.

[END COMMENTS AFTER REBUTTAL]


1. Strengths

As stated in Section 5, "The model presented in this paper is new in the sense that it has never been proposed previously.". In my opinion, the main contribution of this submission is to adapt Synthetic Reduced Nearest Neighbor (SRNN) for regression tasks, while the main contributions claimed in the last part of Section 1 only correspond to some necessary adaptions. SRNN is a traditional method proposed for classification task in 1972 [*], it is reasonable to adapt it for regression task.

[*] Geoffrey W. Gates: The reduced nearest neighbor rule (Corresp.). IEEE Trans. Inf. Theory 18(3): 431-433 (1972)

2. Weaknesses

(1) Motivation:

The original SRNN is proposed mainly for decreasing memory and computation requirements with a possible drop in performance. However, this submission doesn’t mention such motivation or authors have other more reasonable explanations?


(2) Technical details & Experiments:

(a) In experiments, it is shown that the proposed Reg. SRNN achieves superior performance against compared baselines. However, as I mentioned above, SRNN aims at decreasing memory and computation requirements with a possible drop in performance.

(b) The main experimental results are shown in Fig.1&Fig.2. However, in my opinion, it is weird to compare the performance w.r.t. the number of base models (e.g., centroids, RBFs, trees and etc.). For the proposed Reg. SRNN, the number of centroids is just a hyperparameter to be set, and the same to other baselines.

(c) Authors can add the standard kNN model for regression task as a baseline.

(d) Authors analyze the convergence in Subsection 3.7, it is better to illustrate the convergence curve in experiments.


(3) Presentation quality:

(a) The related work should pay more attention on SRNN model rather than some general regression techniques. Besides, this submission is very related to (Tavallali et al., 2020b), so authors can also discuss more about the method in this literature.

(b) The citation style is incorrect.

(c) Section 5 can be merged into Section 6 or Section 1.

(d) What's the difference between Fig.1 and Fig.2. Just different benchmark data sets? If so, they should be merged too.



**Summary Of The Paper:**

The main contributions of this submission is to adapt Synthetic Reduced Nearest Neighbor (SRNN)  for regression tasks. Technical details and comparative studies are provided accordingly.

**Summary Of The Review:**

From my perspective, the motivation is unclear, the experiments are weird and incomplete, and the presentation quality is poor. In summary, I think that this submission is below the standard I expect for an ICLR paper.

---

> ### Author Response · Authors · 2021-11-23
> **Response to Comments of Reviewer 4**
>
> We appreciate your comments and attention to this paper. We will apply your suggestions regarding improving experiments, structure and presentation quality to our next revision of the paper.
>
> Q1: The original SRNN is proposed mainly for decreasing memory and computation requirements with a possible drop in performance. However, this submission doesn’t mention such motivation or authors have other more reasonable explanations?
>
> Ans: It is true that SRNN was originally designed to decrease memory and inference complexity of the model and as result this motivation is also true about this paper. However, we have put more focus on proposing an algorithm that optimizes the model with guarantees on convergence and finding an optimum for the problem.
>
> Q2: (b) The main experimental results are shown in Fig.1&Fig.2. However, in my opinion, it is weird to compare the performance w.r.t. the number of base models (e.g., centroids, RBFs, trees and etc.). For the proposed Reg. SRNN, the number of centroids is just a hyperparameter to be set, and the same to other baselines.
>
> Ans: please look at our response to Q1 to Q3 of the first Reviewer.

---

### Official Review · Reviewer_HzHJ · 2021-11-01

**Correctness:** 3
**Technical Novelty And Significance:** 3
**Empirical Novelty And Significance:** 3
**Recommendation:** 3
**Confidence:** 3

**Main Review:**

This paper proposed a new initialization and EM approach to extend the synthetic reduced nearest neighbor model for regression problems. They have provided theoretical analysis to show the consistency of the proposed method under mild assumptions. Their experimental results show that the proposed method can achieve better or comparable results than existing methods.

The following are my concerns of this paper:
(1) The novelty and contribution of this paper seem limited. The motivation of this paper is to extend the previous method SRNN for regression problems.  The proposed initialization and EM approach seem straightforward based on existing studies.
(2) Some important information is missing in the experiment part. “Various setups for each dataset to achieve the smallest possible train error” is kind of vague. Will the K value in the initialization step have a large influence on the final results?
(3) In Figure 1, what is the difference between SRNN and SRNN0? Figure 1 ‘Slice localization data’ also shows that RF almost has the same curve as bagging. Is there any reason behind this behavior? How are the hyper-parameters tunned for the competing approaches? It mentioned that regboost used trees of depth 3. It would be better also to show the hyper-parameter tunning of the competing approaches.

Minor:
The writing can be further improved.
a local optimum.It --> a local optimum. It
Second step --> The second step
hot-one-vector --> one-hot vector
equation (18) relu --> ReLU if it means the Rectified Linear Unit function.

**Summary Of The Paper:**

This paper proposed a new method to extend the synthetic reduced nearest neighbor model for regression problems. The idea of the proposed method is based on an initialization approach of creating clusters based on the target variable and an EM approach for minimizing the loss function of the proposed Reg. SRNN model. This paper provides an analysis of the consistency of the proposed method. The experiments results compared the proposed methods with several existing approaches.

**Summary Of The Review:**

Please refer to my main review.

---

> ### Author Response · Authors · 2021-11-23
> **Response to Comments of Reviewer 3**
>
> Thank you so much for your comments.
>
> Q1: Some important information is missing in the experiment part. “Various setups for each dataset to achieve the smallest possible train error” is kind of vague. Will the K value in the initialization step have a large influence on the final results?
>
> Ans: Please see the response to Q2 of reviewer 1.
>
> Q2: In Figure 1, what is the difference between SRNN and SRNN0?
>
> Ans: SRNN0 represents the error of the initialization. We will clarify this in the next revision of the paper.

---

### Official Review · Reviewer_c48u · 2021-11-01

**Correctness:** 2
**Technical Novelty And Significance:** 2
**Empirical Novelty And Significance:** 2
**Recommendation:** 3
**Confidence:** 4

**Main Review:**

The description of the results is far from optimal. The problem is the notation, insufficient explanation of main steps, misleading terminology and probably also the correctness:

1) Update step (Section 3.4.2). The problem formulation (15) is clear, however, I didn't get trough the derivation of the approximate problem (18) which is actually used to fined the prototype. More specifically, I don't see the equivalence between the problem (16) and (17). Moreover I think the problems are not equivalent in general. Consider a simple example with two examples and two prototypes in 1D:
- two examples: $x_1=-1,y_1=-1, x_2=1,y_2=1$
- the second prototype $c_2=1, \hat{y}_2-1$
- output of the first prototype $\hat{y}_1$

Then, the minimization (16) w.r.t. $c_1$ is attained for $c_1$ such that $r_{12} < 2$. The minimization of (17) w.r.t. $c_1$ is attend for $c_1$ such that $r_{12} > 2$. Besides, the transition from (17) to (18) needs some justification.

2) The consistency theorem. It is difficult to go through the proof and to understand it.
- "Assume a random point $X_n$ from function $f$ on $X$..." What is a random point from a function?
- What is the meaning of $R_t$ where $t$ is a subset of $X$ ?
- What is a difference between $f(x)$, $\hat{f}(x)$ and $\hat{f}_N(x)$ ?

3) Relation to EM algorithm (section 3.5). I find denoting the proposed optimization algorithm as the expectation-maximization approach misleading. The proposed approach and the EM algorithm are instance of block-coordinate descend like, K-means and many other methods. I do not see any other similarity worth mentioning or which would help to elucidate the problem.

-----
The experiments are not reproducible and poorly described. First, it is not clear which UCI datasets were used and what are the main characteristics of the data (#examples, #dimensions, #outputs). Second, the competing methods are not referenced. Third, there is no discussion of the results at all.

----
Minor issues and typos:
- The notation of the proposed method is no consistent: "Reg-SRNN" and "Reg. SRNN"
- The equations are referenced by a number without brackets .
- Equ (15): $\hat{y}^*_j$ -> $\hat{y}_{j^*_i}$
- Equ (16) and (17): $\min_{c_j}$ -> $\min_{c_k}$
- pp 6: "...sets $S_c$ ..." -> $S_{c_k}$


**Summary Of The Paper:**

The paper proposes a novel algorithm for learning prototype based nearest neighbor regression model. The algorithm minimizes an average of the quadratic loss on training examples w.r.t. the prototype centers and the prototype outputs by a block coordinate descent. The main contribution is the optimization algorithm finding the prototypes.

**Summary Of The Review:**

The paper is in a preliminary state. The explanation of the method is insufficient. The description of experiments lack important information.

---

> ### Author Response · Authors · 2021-11-23
> **Response to Comments of Reviewer 2**
>
> Q1: Update step (Section 3.4.2). The problem formulation (15) is clear, however, I didn't get trough the derivation of the approximate problem (18) which is actually used to fined the prototype. More specifically, I don't see the equivalence between the problem (16) and (17). Moreover I think the problems are not equivalent in general. Consider a simple example with two examples and two prototypes in 1D:
>
> Ans: Thank you so much. You are correct. We noticed that there are two typos in (17,16). First that the optimization should be over c_k and not c_j. Second is that the set of samples specified for each term in 17 are misplaced. The first term, summation must be over S_{c^'_k} and for the second term summation must be over S_{c_k}.
>
> Regarding the transition from 17 to 18, the first term in 18 encourages the samples in s_{c_k} to attract the c_k while the second term encourages the c_k to move away from samples of S_{c^'_k} by at least a distance of r_{ij^{*}} since c_k causes more error to be produced if c_k gets close to samples of S_{c^'_k}.

---

> > ### Comment · Reviewer_c48u · 2021-11-29
> > **Reply to authors.**
> >
> > I tried to correct the formulas (16) and (17) based on your description in the rebuttal. However, I still struggle with the seeing clearly the equivalence between the two problems. Thanks anyway.

---

### Official Review · Reviewer_rWNi · 2021-11-02

**Correctness:** 3
**Technical Novelty And Significance:** 3
**Empirical Novelty And Significance:** Not applicable
**Recommendation:** 5
**Confidence:** 3

**Main Review:**

[COMMENTS AFTER REBUTTAL]

I thank the authors for the rebuttal, I carefully took a look at it, as well as at the reviews of the other reviewers. However, I'm still convinced that it is not strong enough to get accepted in this conference

[END COMMENTS AFTER REBUTTAL]



Strengths:
- the topic is relevant for ICLR
- the idea is simple but potentially interesting
- authors provide also a proof of consistency

Weakness:

W1. I’ve found the structure of the paper not completely clear. In particular I don’t completely understand the order of the subsections in Sect 3: for example, what section 3.2 refers to? Why putting “consistency” before the algorithm and not in the section “Properties of the algorithm” (last subsection)? In Section 3.1 I would also add the description of the SRNN for classification: the proposed approach is an extension of it, thus it would be beneficial to summarize it.


W2. The section 2 “Related Work” contains a general introduction to the regression problem. I think this is too general, and not relevant for the paper. On the other hand, it does not contain the description of works which are more strictly related to the presented method, such as the general nearest neighbour strategies and especially the method in Huang (2017): these have been only cited at the end of the section. In particular I think it is fundamental to define the differences between the proposed approach and that of Huang (2017) – authors only reported a general sentence (“However, the proposed algorithm does not have guarantee of convergence or achieving some sort of optimum solution.”), without describing the algorithm and the idea. Finally, I suggest authors to clearly discuss the differences with the work of Tavallali et al 2020b, which has been cited in many different places inside the manuscript.


W3. The authors use a K-means like strategy. Many methods in the literature used such generalized strategies, I think authors should at least acknowledge some of them. Moreover, it is clear from  such methods that the initialization is always an issue. How did authors solve it? With the procedure described in Section 3.2? If so, how did they initialize the k-means inside it? How many repetitions? Which is the impact of this procedure (also empirically)?
Please see also the discussion in:

P Fränti, S Sieranoja: How much can k-means be improved by using better initialization and repeats? Pattern Recognition, 2019



W4. How did author face the problem of determining the number of centroids K? How relevant is this choice? I suggest the authors to at least discuss this crucial issue


W5. I think the experimental part can be largely improved. Here are some comments:
1. Many important details are missing. For example:
- Details on datasets: number and (at least) names of the datasets, characteristics in terms of number of features, number of objects, dimensions of training and testing sets (if any) and so on
- Details on protocols: evaluation metric, details on Cross Val (e.g. how many repetitions of 80%/20%?)
- Details on the proposed approach: initialization, number of iterations, different options investigated, impact on the performances and so on. For example,  authors used K=4 for one dataset and K=2 for the others: what if we change this value? How critical is this choice?
- What is the meaning of “We used various setups for each dataset to achieve the smallest possible
train error.”? Which setups? Is this behaviour fair? Please add a justification


2. For what concerns the chosen competitors: instead of using these general methods, I suggest the authors to add a comparison with some of the original NN regression methods and with the method of Huang, which represents the most similar approach in the literature.

3. Why authors consider the different clusters in their approach as “base models” (like trees in forests?)

4. Plots reproduced in fig 1 are very difficult to read. Even zooming the pdf their presentation remains obscure (e.g. the colors of the different competitors are too similar)

5. Comments on the results are missing: there is only one very general sentence (“As can be observed from figures 1, and 2, the SRNN-Reg was able to achieve better or comparable train and test errors to other models.”). Example of aspects which can be discussed are:
+ how does the method work in the different datasets? Any relation with the different characteristics?
+ which are the best competitors?
+ why in some plots we only have 6 or 7 lines?
+ why in some plots we have horizontal lines?



**Summary Of The Paper:**

The paper presents a novel regression model based on Nearest Neighbor. In particular authors adapt the synthetic reduced NN model used for classification to the regression case. The main intuition is to simplify the regression problem by finding clusters of inputs and targets. Authors presents some theoretical properties of the method as well as few experimental results.

**Summary Of The Review:**

Potentially interesting paper with some problems in the presentation and in the experiments.

---

> ### Author Response · Authors · 2021-11-23
> **Response to Comments of Reviewer 1**
>
> Thank you so much for your comments regarding the structure of the paper and its organization. We will address these issues in the next version by changing the related work and decreasing the general related work regarding regression, discussing more related SRNN papers and putting the consistency section after the proposed method.
>
> In the following, we will respond to rest of the questions:
>
> Q1 (W3): The authors use a K-means like strategy. Many methods in the literature used such generalized strategies, I think authors should at least acknowledge some of them. Moreover, it is clear from such methods that the initialization is always an issue. How did authors solve it? With the procedure described in Section 3.2? If so, how did they initialize the k-means inside it? How many repetitions? Which is the impact of this procedure (also empirically)?
>
> Ans: Regarding the Regression SRNN, in our search we did not find any paper that specifically discusses the initialization for Regression SRNN. However, the initialization for classification version of it already existed in the literature and was explained previously.
>
> Q2 (W4): How did author face the problem of determining the number of centroids K? How relevant is this choice? I suggest the authors to at least discuss this crucial issue
>
> Ans: Our problem here was not to determine K. It was rather coming up with an optimization algorithm that is capable of optimizing the model that has guarantees, can minimize the error of this problem (which is not properly addressed in the literature) and also perform competative to other existing similar models. However, in practice, one can determine K through cross-validation or K-fold cross validation. Here we have demonstrated that the algorithm can optimize the model with various values of K and can achieve comparable or better performance compared to other models.
>
> Q3: Details on the proposed approach: initialization, number of iterations, different options investigated, impact on the performances and so on. For example, authors used K=4 for one dataset and K=2 for the others: what if we change this value? How critical is this choice?
>
> Ans: Empirically, we noticed that usually the choice of K=2 or K=4 for the number of output clusters for the intialization results in an intialization with lower train error.
>
> Q4: What is the meaning of “We used various setups for each dataset to achieve the smallest possible train error.”? Which setups? Is this behaviour fair? Please add a justification
>
> Ans: thank you for this question. We meant the setup of learning step sizes in the optimization of centroid step.
>
> Q5: Why authors consider the different clusters in their approach as “base models” (like trees in forests?)
>
> Ans: The model can be seen as a kind of ensemble where only one baseline model produces the output prediction. In that sense the model is similar to ensemble models. This provides a logical basis for a fair comparison with other models. We have tried to compare models in a sense that they have similar number parameters or number of base models. E.g., a tree is comparable with the model in the sense that it creates partitions in the input space but the number of parameters are not comparable. On the other hand, a forest with similar number of parameters (but similar number of base line trees) can be a better model for comparison since it has more prediction power.
>
> Q6: how does the method work in the different datasets? Any relation with the different characteristics?
> which are the best competitors?
> why in some plots we only have 6 or 7 lines?
> why in some plots we have horizontal lines?
>
> Ans: Currently we do not have any specification on the datasets that the model works better with. However, any dataset that the samples are is produced from multiple normal distributions can be handled properly with the model since the model is originally designed for such datasets.
>
> Again, each model can handle a dataset differently but we have noticed that forest and ensemble models are better competitors to Reg. SRNN.
>
> In some plots the poorly performing models are not presented simply because they have performed so poorly that their curve falls outside the boundaries of the figure.
>
> The horizontal lines represent linear and ridge regression.

---

### Decision · Program_Chairs · 2022-01-20

**Decision:**

Reject

**Comment:**

An algorithm for learning prototype based nearest neighbor regression model is presented. This algorithm minimizes an MSE on training examples w.r.t. the prototype centers and the prototype outputs by a block coordinate descent. The main contribution is the optimization algorithm finding the prototypes.
Major concerns in the reviews include missing mathematical rigor, poor description of the experiments, and unclear novelty. From my own reading I would like to add that the main theoretical contribution (Theorem 1) makes assumptions that are beyond any reasonable constraint, in particular as we know for more than 40 years, that such kind of assumptions are superfluous for many, many other algorithms.

In summary, a clear reject.